# Inflammation and Cognition in Bipolar Disorder: Diverging Paths of Interleukin-6 and Outcomes

**DOI:** 10.3390/ijms26136372

**Published:** 2025-07-02

**Authors:** Ulises Ríos, Susana Pérez, Constanza Martínez, Pablo R. Moya, Marcelo Arancibia

**Affiliations:** 1Departamento de Psiquiatría, Escuela de Medicina, Facultad de Medicina, Universidad de Valparaíso, Valparaíso 2360002, Chile; ulises.rios@uv.cl (U.R.); susana.spr@gmail.com (S.P.); constanza.csy@gmail.com (C.M.); 2Centro de Estudios Traslacionales en Estrés y Salud Mental (C-ESTRES), Universidad de Valparaíso, Valparaíso 2360102, Chile; pablo.moya@uv.cl; 3Instituto de Fisiología, Facultad de Ciencias, Universidad de Valparaíso, Valparaíso 2360102, Chile

**Keywords:** bipolar disorder, inflammation, cognition, interleukin-6, C-reactive protein, hospitalization, suicide, mood disorders, mental disorders

## Abstract

Bipolar disorder (BD) may present with neurocognitive dysfunction due to inflammatory alterations through different biological pathways. However, findings are not consistent regarding the patterns of neurocognitive dysfunction and elevation of inflammatory biomarkers during the different mood phases. Therefore, we aimed to determine associations between inflammatory biomarkers, neurocognitive functioning, and clinical outcomes in patients with BD in euthymia. We conducted a cross-sectional study including 109 adults. Serum levels of interleukin-6 (IL-6), high-sensitivity C-reactive protein (hs-CRP), neurocognitive parameters (ACER), number of suicide attempts (SA), and hospitalizations (NH) were measured. We found negative and moderate correlations between IL-6 and ACER total score, language, visuospatial abilities, and orientation/attention. There was a positive and moderate correlation between IL-6 and NH. IL-6 significantly predicted ACER total score, language, memory, orientation/attention, visuospatial abilities, and NH. Overall, IL-6 had an inverse association with neurocognition and clinical variables, whereas hs-CRP did not play a role. Here we demonstrate that IL-6 predicts neurocognitive functioning in adults with BD. BD may be a biological model for studying the relationship between inflammation and neurocognition in severe psychiatric disorders. Prospective studies at different mood phases of the disease must be conducted.

## 1. Introduction

Bipolar disorder (BD) is a chronic disorder characterized by recurrent mood fluctuations between the poles of depression and mania [1]. It was one of the first psychiatric illnesses to receive attention from areas such as genetics and biology due to the clinical observation that the illness runs in families [2]. In this regard, inflammation has been found to play a cross-cutting role in severe psychiatric disorders, especially BD. To approach the study of the biological basis of multiple neuropsychiatric conditions, endophenotypes have acquired great relevance. This concept designates biomarkers of different nature that are measurable, that have a hereditary component, and that represent the underlying biological mechanisms of neuropsychiatric disorders [3]. Two proposed endophenotypes in BD are neurocognitive and inflammatory alterations [3]. In turn, both could be closely related.

Neurocognition refers to cognitive functions that are closely linked to specific areas, neural pathways, or networks in the brain. It determines multiple clinical outcomes and is a key prognostic factor in severe psychiatric disorders [4]. In BD, neurocognitive disturbances are frequent even in euthymia [5], which disrupt an individual’s ability to live independently. They include alterations in executive functions, processing speed, attention, memory, and social cognition [6,7,8]. Douglas et al. [9] found that 64% of patients with BD during a depressive episode and 57% of euthymic individuals were impaired in one or more cognitive domains. A systematic review including five studies concluded that participants with BD in euthymia showed greater cognitive dysfunction than those with mild cognitive impairment, but less than those with dementia [8]. However, a meta-analysis of longitudinal studies of patients with BD did not corroborate progressive cognitive impairment over a 5-year period [6]. In addition, a 5-year longitudinal study described the trajectories of change in the cognitive profile of people with BD with low, medium, and high cognitive performance. In none of the three groups was there a decline, nor were there significant differences between these trajectories [7]. At baseline, only a minority of participants had low cognitive performance, which was associated with a greater presence of childhood trauma and more psychiatric hospitalizations. These findings do not support the neuroprogression hypothesis in BD, as other authors have stated [10,11,12]. However, they do support the idea that the greater the allostatic load, the greater the cognitive dysfunction. There are different hypotheses that seek to explain how neurocognitive alterations originate in BD. At the neurobiological level, in both unipolar and bipolar mood disorders, morphofunctional changes have been found in brain regions involved in mood regulation and cognitive functioning, including the orbitofrontal cortex, amygdala, and hippocampus [13]. Some research groups have proposed that inflammation is the biological substrate of the brain alterations that account for cognitive dysfunction [14,15].

Although the inflammatory theory of mental disorders has occupied a central place in research on the biological basis of severe psychiatric disorders, its relationship to neurocognitive dysfunction is incipient. Recently, the GWAS by Mullins et al. [16] revealed for the first time that the locus associated with the major histocompatibility complex had one of the most significant associations with BD, superior to those loci linked to the mechanisms classically described for the disease. In this line, there are different primary studies that have examined the association between inflammatory biomarkers and some clinical outcomes in BD. Some have concluded that there is a multisystemic inflammation from the early stages of the disease [17]. Two of the most studied inflammatory biomarkers in BD are C-reactive protein (CRP) and interleukin-6 (IL-6). CRP is an acute-phase protein that increases its levels in inflammatory contexts and is induced by IL-6 [18]. High-sensitivity CRP (hs-CRP) is a standardized way of representing the low-grade inflammatory response observed in BD, whose magnitude is greater than in healthy controls, an observation verified by primary studies [19,20] and systematic reviews [21,22,23]. However, this elevation is not a consistent finding in all patients with BD [24] nor in all mood phases of the condition [25,26,27].

It has been proposed that the activation of the HPA axis during manic episodes would generate an initial increase in inflammatory parameters [25]. From a pathophysiological point of view, the increase in CRP promotes the permeability of the blood–brain barrier, allowing the diffusion of proinflammatory cytokines and antibodies that morphofunctionally alter central structures [28], favoring the production of free radicals and altering mitochondrial function, neurotransmitter synthesis, glial function, synaptic remodeling, and neurogenesis [29,30]. Regarding IL-6 levels, a systematic review with meta-analysis found that this biomarker was significantly higher in patients with BD compared to controls, both in mania and euthymia [31]. Consistently, another systematic review concluded that IL-6 remains elevated regardless of the mood phase, indicating that it is a potential biomarker of the disease [23].

In healthy elderly people, increased levels of CRP and IL-6 have been found to be associated with volumetric reductions in hippocampal gray matter [32], structural alterations of the white matter [32], and cortical atrophy [33]. Moreover, inflammatory biomarker levels are associated with cognitive performance in people with major depressive disorder [34,35]. In BD, multiple mood episodes alter the homeostasis between inflammatory mechanisms, oxidative processes, and neuroprotective mechanisms, promoting neuronal apoptosis, which increases the individual’s vulnerability to psychological stress and cognitive impairment [36]. However, the association between inflammatory parameters and cognitive dysfunction has been scarcely studied in samples with BD [35]. Likewise, the potential influence that inflammatory states have on clinical outcomes that account for the severity of the condition, such as suicidality and the number of hospitalizations, is uncertain. The aim of this study is to analyze the association between inflammatory biomarkers, neurocognitive functioning, and clinical outcomes in patients with BD in euthymia. As a hypothesis, we propose that higher levels of inflammatory biomarkers will be related to worse general and specific cognitive functioning and worse clinical outcomes.

## 2. Results

### 2.1. Descriptive Analysis

The sample included 109 participants, of whom 72 were women (66%). The mean age of the subjects was 47.4 ± 14 years. Table 1 presents the levels of inflammatory markers, ACER scores, and clinical variables.

### 2.2. Bivariate Analysis

Significant, negative, and moderate correlations were found between IL-6 levels and ACER-T (r = −0.326; *p* = 0.0007), ACER-L (r = 0.307; *p* = 0.0015), ACER-VE (r = −0.322; *p* = 0.0008), and ACER-OA (r = −0.312; *p* = 0.0011). There was a significant, negative, weak correlation between IL-6 and ACER-M levels (r = −0.236; *p* = 0.0146). Also, IL-6 had a significant, positive, moderate correlation with NH (r = 0.399; *p* < 0.001). There were no significant correlations involving hs-CRP levels. Significant correlations were found between ACER-T, ACER-L, ACER-VE, and ACER-VF with clinical outcomes (Table 2).

When comparing the high and low neurocognitive performance groups, categorized according to the sample mean ACER-T (85.7 ± 9.2), no significant differences were found in hs-CRP or IL-6 levels (Table 3).

Conversely, when comparing categorizing according to high or low levels of inflammation using the mean IL-6 (8.3 ± 28.4) as a cut-off point, no differences were found in neurocognition, but the NH was significantly higher in the group with high IL-6 levels (2.9 vs. 7.3; *p* = 0.001) (Table 4).

There were no differences in ACER scores or clinical outcomes when comparing subjects according to hs-CRP levels above or below the mean (0.2 ± 0.4) (Table 5).

### 2.3. Linear Regression Models

Several regression models were analyzed to explore the influence of IL-6 and hs-CRP levels on neurocognitive functioning and clinical outcomes. Significant models were found to predict levels for ACER-T, ACER-L, ACER-VE, ACER-OA, ACER-M, and NH. Overall, inflammatory variables explained about 10% of the variation in neurocognitive dimensions and NH. IL-6 level was the significant variable in the models that reached statistical significance. An increase in IL-6 values was associated with a decrease in ACER-T, ACER-L, ACER-VE, ACER-OA, and ACER-M, and an increase in the NH (Table 6).

Taken together, our results show that the sample had high levels of IL-6 and normal levels of hs-CRP. The group with higher-than-average IL-6 levels had a higher SA. This trend was corroborated in the correlation analysis, where the higher the IL-6 levels, the higher the average SA. Conversely, higher IL-6 levels correlated with worse neurocognitive indicators. Indeed, regression models verified that IL-6 levels predicted neurocognitive functioning and NH.

## 3. Discussion

Our study analyzed the associations between inflammatory biomarkers, neurocognitive functioning, and clinical outcomes in patients with BD type I in euthymia. Significant negative correlations were found between IL-6 levels and neurocognitive functioning, and positive correlations between IL-6 and clinical variables as NH. IL-6 levels significantly predicted total neurocognitive functioning and the dimensions of language, memory, orientation/attention, and visuospatial abilities.

There is sufficient evidence linking inflammation with neurocognitive alterations in mood disorders [36,37], particularly in major depressive disorder [34,38]. However, studies focused on BD are much scarcer, especially those analyzing IL-6 as an inflammatory biomarker. Also, most of the published evidence comes from cross-sectional designs, so they cannot evaluate causal hypotheses that clarify the temporal order of the biological phenomena, that is, whether an inflammatory state precedes a major mood disorder or whether it is this alteration that promotes inflammation.

A theoretical line of research suggests a potential association between IL-6 and cognitive dysfunction in patients with BD in euthymia. This association would indicate that persistently elevated levels of IL-6 generate an inflammatory environment that structurally alters specific brain regions, with an inverse correlation between peripheral inflammatory markers and brain region volume [39,40], which is associated with cognitive deficits [36,41]. In people with BD, some of the areas most sensitive to inflammation are the anterior cingulate cortex, amygdala, prefrontal cortex, hippocampus, and orbitofrontal cortex [40], all involved in the pathophysiology of BD. At the molecular level, IL-6 can inhibit cortical neurotransmitter release through a direct action on the presynaptic neuron [42]. There is also evidence showing that multiple mood episodes alter the homeostasis between inflammatory mechanisms, oxidative processes, and neuroprotective mechanisms, resulting in cognitive alterations [36].

In our study, elevated IL-6 (8.39 ± 28.4) and normal hs-CRP levels (0.24 ± 0.44) were found. This conclusion is variable in the published evidence, since it has been pointed out that CRP levels are higher in subjects with BD [21,22,23] or show no differences with the general population [43]. The systematic review by Bauer et al. [36] found a negative correlation between CRP levels and performance in language, memory, and attention tests in a sample of 107 people with BD. We found no significant correlations among ACER scores and hs-CRP. This lack of association may be partially explained due to the normal hs-CRP levels in our sample, but it is important to consider that IL-6 directly reflects the activity of cytokines, which are the main mediators and regulators of the inflammatory response, while hs-CRP is a more global marker of the acute-phase response, but does not indicate the specific cytokine or molecular pathways involved. In this line, the average IL-6 had a large dispersion in our sample. This may be due to different phenomena beyond psychiatric diagnosis, such as age, sex, genetic variants involved in IL-6 signaling, and concomitant diseases, among others [44]. These factors were not considered in the adjustment of the proposed models, a fact that constitutes a limitation of this study.

IL-6 is a pleiotropic cytokine capable of exerting divergent effects on cellular metabolism [45] and, specifically, on the immune system [46]. The most recently published GWAS of BD highlighted that the main hit was a variant of the gene encoding for the major histocompatibility complex [16]. In addition, individuals with major depressive disorder and BD have alterations of the compensatory immune-regulatory reflex system, a mechanism that regulates the primary immune response to contribute to recovery from the acute phase of the illness; among the alterations found in the reflex system are abnormalities of classical IL-6 signaling [47]. Based on this evidence and the results showing inflammatory alterations in patients with BD in euthymia [23,31,48], it could be suggested that a sustained low-grade inflammatory state may configure an endophenotype of the disease. Our results seem to support this hypothesis. In this line, some authors affirm that neuroinflammation constitutes an endophenotype in BD [3] and, more specifically, IL-6 levels [49]. From a clinical point of view, the immunomodulatory effect of mood stabilizers such as lithium [50] and lamotrigine [51], frequently used in the management of BD, is known. From a cellular perspective, inflammatory states contribute to telomere shortening and thus to aging, characterized by cognitive decline; in the opposite direction, telomeric shortening can induce low-grade inflammatory states, favoring a bidirectional process [52]. However, both telomeric shortening and an increase in inflammatory parameters in patients with BD are not consistently verified in unaffected first-degree relatives [53], so their validity as an endophenotype should be further studied.

The findings of this study demonstrate that an inflammatory state, as indicated by elevated IL-6 levels, significantly predicts a decrease in ACER-T values and the domains of memory, language, visuospatial abilities, and attention/orientation. These findings are consistent with those reported by Wiener et al. [54], where higher IL-6 levels correlated with more severe cognitive dysfunction in a sample of patients with BD. In the study by Barbosa et al. [55], IL-6 levels were found to significantly predict cognitive functioning in patients with BD in euthymia. In a similar study, Poletti et al. [56] employed an elastic net penalized regression on a sample of patients with BD; their findings indicated that higher IL-6 levels were associated with an increased likelihood of having poor cognitive performance. As mentioned above, the proposed mechanisms by which IL-6 levels alter neurocognitive functioning are multiple. Recently, studies of resting-state functional connectivity magnetic resonance imaging have suggested that immune dysregulation is involved in connectivity abnormalities in limbic and somatomotor networks that account for the neurocognitive alterations in BD. These alterations include structural modifications, such as reduced cortical thickness and subcortical volumes [18]. At the cellular level, IL-6 is known to modulate the process of neurogenesis throughout neurodevelopment. However, prolonged exposure at high levels may lead to reduced proliferation and increased neuronal apoptosis. This would be one of the main mechanisms underlying the alterations that occur in the aforementioned structures, all of which are involved in cognitive processes. However, the conclusions on the effects of IL-6 at the cellular level have been obtained from in vitro studies and animal models; therefore, it is necessary to deepen research in this area [57].

The published evidence analyzing the effect of inflammatory status on clinical outcomes in BD is even scarcer. The systematic review and meta-analysis by Miola et al. [58] included 21 studies and 7682 participants with a psychiatric diagnosis (7445 with mood disorders), finding a large association between CRP concentrations and suicidal ideation (SMD 1.145, 95% CI 0.273–2.018), and a medium association with suicide attempts (SMD 0.549, 95% CI 0.363–0.735). Our results did not corroborate a relevant role of hs-CRP in the prediction of SA, nor of neurocognitive functioning. On the other hand, IL-6 levels have received more attention as a biomarker of suicidal behavior [59]. Indeed, the meta-analysis by González-Castro et al. [60] (n = nine studies) showed that suicidal behaviors were associated with higher levels of IL-6 in serum and cerebrospinal fluid, while the meta-regression analysis indicated an association between plasma and serum IL-6 levels with male sex. These conclusions are not concordant with our findings, since, in our study, IL-6 levels were significantly associated with a higher NH but not SA. However, it is important to consider that the results of Miola et al. [58] and González-Castro et al. [60] incorporate samples with a wide variety of psychiatric diagnoses, so they are not directly comparable with our sample. Inflammatory burden has been clearly recognized as a preponderant factor in immune-mediated diseases [61], but in psychiatric conditions, this relationship has been scarcely studied. In unipolar depression, a correlation between inflammatory burden and depressive outcomes has been corroborated [62], including hospital variables such as those observed in our results. In BD, Queissner et al. [63] concluded that individuals who were exposed to a higher level of inflammation over time suffered from more manic symptoms in this period. This finding allows us to indirectly hypothesize that people with higher levels of inflammation would require a greater NH. However, the relationship between inflammation and NH is still statistical and requires further theoretical understanding. Despite its biological plausibility, the clinical utility of pharmacological targets that seek to modulate the IL-6-mediated inflammatory pathway is a matter of debate. To date, there are few clinical trials with highly variable results regarding the pro-cognitive effect of IL-6 modulators; most of these studies have been conducted in samples of patients with Alzheimer’s disease [64], making direct extrapolation to our population of interest difficult. In parallel, other interventions of a non-pharmacological nature, such as dietary interventions, have received some degree of interest as modulators of the IL-6-related inflammatory pathway, but evidence is still scarce [65].

Among the limitations of this study, its cross-sectional design does not allow us to establish a causal hypothesis regarding the phenomena studied. The sample size was limited, and there was a disparity between the proportion of men and women. We did not incorporate a control group constituted by participants without BD, an aspect to be considered in future designs; however, we focused on the concept of “intra-group differences,” emphasizing the analysis of the clinical heterogeneity that results from the classification based on the traditional diagnostic categories. Regarding the measurement of neurocognition, ACER offers a first global assessment, so other instruments can complement a more in-depth evaluation of this construct. Additionally, there is great heterogeneity in the instruments used to measure neurocognition, which makes it difficult to compare our own results. We also did not have measurements of inflammation and cognition in active mood phases of the disease, nor did we control for factors that may have modified the results, such as the use of antipsychotics, the presence of chronic diseases related to inflammation, and body weight. On the other hand, the regression models explained only a small proportion of the variance of the cognitive and clinical outcomes, which could be since other relevant variables were not considered, especially clinical variables. Nevertheless, there is sufficient evidence from in vitro analyses, in vivo models, clinical studies, and meta-analyses to support that inflammation has a significant impact on clinically meaningful outcomes. Our results allow a detailed analysis of different dimensions of neurocognition and its relationship with inflammatory biomarkers, not always present in the published evidence, where neurocognition is mostly reported as a global outcome.

## 4. Materials and Methods

### 4.1. Design

We developed an exploratory, analytical, cross-sectional study from the Ríos cohort [66]. Our study presents a methodological design similar to that used in other studies with comparable objectives and samples [20,67,68].

### 4.2. Participants

We included adults (18 to 65 years old) with a diagnosis of BD type I according to DSM-IV-TR [69] in euthymia for three months or more, assessed with the Young Mania Rating Scale (≤6 points) [70] and the Hamilton Depression Rating Scale (≤6 points) [71]. Subjects with active substance use or who had received electroconvulsive therapy during the previous three months were excluded. We did not exclude patients with medical comorbidities or use of specific drugs.

### 4.3. Instruments

The Addenbrooke’s Cognitive Examination Revised (ACER) was administered to assess neurocognitive performance. This instrument is designed to detect different types of neurocognitive disorders, and its performance would not be affected by mood symptoms or mood disorders. It evaluates five cognitive domains: orientation/attention (18 points), memory (26 points), verbal fluency (14 points), language (26 points), and visuospatial abilities (16 points). We used the version of the instrument validated in the Chilean population [72].

### 4.4. Clinical Variables

Three clinical variables of relevance in the course of BD were analyzed: age of disease onset (according to age at diagnosis), number of suicide attempts (SA), and number of hospitalizations (NH). Data were collected from the clinical records of each patient.

### 4.5. Inflammatory Variables

Inflammatory analyses were conducted at the Toxicology Laboratory Universidad de Valparaíso. Quantitative determination of IL-6 and hs-CRP in serum was performed using peripheral venous blood samples (forearm vein) by standard venipuncture. All blood samples were collected between 07:00 and 09:00 A.M. into vacuum blood-collecting tubes containing 1 mg/mL of anticoagulant ethylenediaminetetraacetic acid (EDTA), and the samples were obtained by centrifugation immediately for 10 min at 3500 rpm.

#### 4.5.1. hs-CRP

The determination and quantification of hs-CRP was performed using BN ProSpec^®^ (Erlangen, Germany) automated equipment, and the reagent used for this measurement was CardioPhase^®^ (Erlangen, Germany) hs-CRP. The technique used was immunonephelometry with intensifying particles, which consists of the use of polystyrene particles coated with a specific monoclonal antibody against human CRP, which, when mixed with samples containing CRP, form aggregates that scatter the incident light beam. The intensity of the scattered light depends on the concentration of the corresponding protein in the sample. The final titration is made by comparison with a standard curve of known concentration, with the lower limit of this curve being the sensitivity (LOD) of the assay, which typically corresponds to 0.0175 mg/dL. The following reference values were used according to the manufacturer’s instructions for CardioPhase^®^ hs-CRP with BN ProSpec^®^ equipment: <0.10 mg/L (low risk), 0.10–0.30 mg/L (moderate risk), >0.30 mg/L (high risk), and >1.00 mg/L (very high risk).

#### 4.5.2. IL-6

Quantitative determination of IL-6 was performed through a quantitative enzyme-linked immunosorbent assay (ELISA) using a commercially available kit (Quantikine, R&D Systems, Minneapolis, MN, USA), taking IL-6 < 3.5 pg/mL as the reference value according to the manufacturers’ instructions. Briefly, this assay employs the quantitative sandwich enzyme immunoassay technique, in which the microplate is pre-coated with a monoclonal antibody specific for human IL-6. A total of 100 μL of standards or the plasma sample was added to each well and incubated for 2 h at RT, and any IL-6 present was bound by the immobilized antibody. To construct a calibration curve, we diluted Human IL-6 Standard with the 1× Assay diluent to 300, 100, 50, 25, 12.5, 6.25, 3.13, and 0 pg/mL and used them as standard samples. After washing away any unbound substances, 200 μL of human IL-6 conjugate (enzyme-linked polyclonal antibody specific for human IL-6) was added to the wells, followed by a 2 h incubation at room temperature. Then, 200 μL of substrate solution was added to each well and incubated for 20 min at RT, with the color developing in proportion to the amount of IL-6 bound. The color development is stopped with 50 μL of the stop solution, and the intensity of the color is measured. Absorbance was interpolated from a calibration curve. The sensitivity was 0.7 pg/mL in a range of 0–300 pg/mL with intra-assay and inter-assay coefficients from 4.2% to 7%, respectively. There were no undetectable values. All samples were over the detection limit [73].

### 4.6. Statistical Analysis

For descriptive statistics, absolute numbers, proportions, and means (standard deviations) were used. For inferential statistics, a correlation matrix was performed to analyze the associations between IL-6, hs-CRP, ACER scores, and clinical variables (Pearson correlation test). IL-6 and hs-CRP means were compared according to high or low neurocognitive performance categorized according to the ACER-T mean (Student’s *t*-test); likewise, the means of ACER scores were compared according to groups divided by values below or above the mean of IL-6 and hs-CRP. Multiple linear regression models were performed to explore the influence of inflammation variables on neurocognitive performance and clinical outcomes. A significance level of 5% was used. Data were analyzed in Stata 17 (Statacorp, College Station, TX, USA).

## 5. Conclusions

The findings of this study support the hypothesis that IL-6 levels correlate inversely with cognitive functioning in patients with euthymic BD, allowing prediction of dimensions such as memory, language, visuospatial abilities, and orientation/attention. In consideration of the clinical outcomes, elevated IL-6 levels were associated with an increase in the NH (Figure 1). Different mechanisms related to the effect of I-L6 on central brain structures could be involved, such as the inflammatory effect on telomere structure, dysfunction at the synaptic level, activation of apoptotic pathways, and morphofunctional impact on specific regions. All these mechanisms have been linked to the pathophysiology of BD. Nevertheless, BD poses the challenge of studying biological mechanisms by means of biomarkers of a different nature from a fluctuating perspective according to the mood phase considered. Studies employing prospective designs, incorporating neurocognition and inflammation measurements from an evolutionary perspective, and at different mood stages of the disease, will help to elucidate the causality hypotheses discussed in this article. More importantly, these studies will enable the development of more precise interventions that extend beyond the recovery from euthymia.

## Figures and Tables

**Figure 1 ijms-26-06372-f001:**
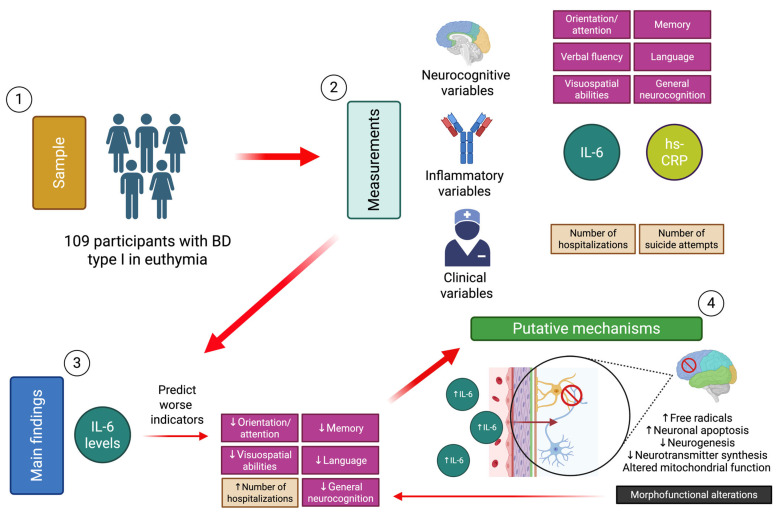
Synthesis of the study methods, main findings, and proposed mechanisms [74]. Down arrows symbolize a decrease and up arrows an increase.

**Table 1 ijms-26-06372-t001:** Levels of inflammatory biomarkers, ACER scores, and clinical variables.

Variable	Mean	Standard Deviation
IL-6 (pg/mL)	8.3	±28.4
hs-CRP (mg/L)	0.2	±0.4
ACER-T	85.7	±9.2
ACER-L	24.1	±1.9
ACER-VE	14.3	±1.9
ACER-OA	16.6	±1.7
ACER-M	20	±4.4
ACER-VF	10.5	±2.7
Age of onset (age)	23.4	±9.3
Time of evolution (years)	23.9	±12.9
NH	3.3	±3.9
SA	1.1	±2

IL-6: interleukin-6; hs-CRP: high-sensitivity C-reactive protein; ACER-T: total; ACER-L: language; ACER-VE: visuospatial abilities; ACER-OA: orientation/attention; ACER-M: memory; ACER-VF: verbal fluency; NH: number of hospitalizations; SA: number of suicide attempts.

**Table 2 ijms-26-06372-t002:** Correlation between inflammatory biomarkers, ACER scores, and clinical variables.

	hs-PCR	IL-6	ACER-T	ACER-L	ACER-VE	ACER-OA	ACER-M	ACER-VF	Age	Onset	TE	NH
IL-6	0.05 (0.5946)											
ACER-T	0.076 (0.4510)	−0.326 * (0.0007)										
ACER-L	0.143 (0.1543)	−0.307 * (0.0015)	0.645 * (<0.001)									
ACER-VE	0.062 (0.5364)	−0.322 * (0.0008)	0.513 * (<0.001)	0.357 * (0.0001)								
ACER-OA	0.029 (0.7667)	−0.312 * (0.0011)	0.554 * (<0.001)	0.253 * (0.0077)	0.260 (0.0060)							
ACER-M	0.059 (0.5529)	−0.236 * (0.0146)	0.852 * (<0.001)	0.422 * (<0.001)	0.221 * (0.0203)	0.299 * (0.0014)						
ACER-VF	−0.005 (0.9542)	−0.058 (0.5495)	0.755 * (<0.001)	0.342 * (0.0003)	0.206 * (0.0302)	0.342 * (0.0002)	0.546 * (<0.001)					
Age	0.116 (0.2453)	0.117 (0.2362)	−0.277 * (0.0038)	−0.176 (0.0685)	−0.283 * (0.0029)	−0.127 (0.1856)	−0.092 (0.3407)	−0.366 * (0.0001)				
Onset	0.094 (0.3456)	0.043 (0.6601)	−0.027 (0.7752)	0.101 (0.2976)	−0.168 (0.0822)	−0.113 (0.2407)	0.076 (0.4278)	−0.101 (0.2929)	0.446 * (<0.001)			
TE	0.057 (0.5667)	0.095 (0.3368)	−0.280 * (0.0035)	−0.263 * (0.0060)	−0.186 (0.0530)	−0.057 (0.5538)	−0.155 (0.1069)	−0.325 * (0.0006)	0.765 * (<0.001)	−0.233 * (0.0144)		
NH	−0.021 (0.8311)	0.399 * (<0.001)	−0.235 * (0.0147)	−0.246 * (0.0106)	−0.072 (0.4538)	−0.016 (0.8684)	−0.169 (0.0776)	−0.269 * (0.0046)	0.206 * (0.0309)	−0.077 (0.4255)	0.280 * (0.0032)	
SA	0.03 (0.7610)	−0.046 (0.6405)	0.066 (0.4961)	0.041 (0.6695)	0.152 (0.1146)	−0.152 (0.1126)	0.138 (0.1498)	−0.048 (0.6184)	−0.113 (0.2417)	−0.259 * (0.0065)	0.063 (0.5116)	0.025 (0.7949)

Coefficient correlations (*p*-Value). Statistically significant correlations are marked with an asterisk. IL-6: interleukin-6; hs-CRP: high-sensitivity C-reactive protein; ACER-T: ACER total; ACER-L: language; ACER-VE: visuospatial abilities; ACER-OA: orientation/attention; ACER-M: memory; ACER-VF: verbal fluency; Age: patient’s age; Onset: age of onset; TE: time of evolution; NH: number of hospitalizations; SA: number of suicide attempts.

**Table 3 ijms-26-06372-t003:** Comparison of inflammatory biomarker levels according to ACER-T.

	Low ACER-T (n = 43)	High ACER-T (n = 66)	*p*-Value
IL-6	10.2 ± 40.5	7.3 ± 16.6	0.61
hs-CRP	0.22 ± 0.26	0.25 ± 0.53	0.79

IL-6: interleukin-6; hs-CRP: high-sensitivity C-reactive protein; ACER-T: total.

**Table 4 ijms-26-06372-t004:** Comparison of ACER scores and clinical outcomes according to IL-6 level.

	Low IL-6 (n = 96)	High IL-6 (n = 10)	*p*-Value
ACER-T	85.8 ± 8.6	83.3 ± 14.4	0.42
ACER-L	24.12 ± 1.7	24.6 ± 3	0.46
ACER-VE	14.36 ± 1.87	13.7 ± 2.86	0.31
ACER-OA	16.62 ± 1.7	16.40 ± 2.5	0.7
ACER-M	20.12 ± 4.42	18.7 ± 5.69	0.34
ACER-VF	10.59 ± 2.79	9.9 ± 2.99	0.45
NH	2.97 ± 3.36	7.3 ± 7	0.001 *
SA	1.25 ± 2.11	0.9 ± 1.59	0.6

IL-6: interleukin-6; hs-CRP: high-sensitivity C-reactive protein; ACER-T: total; ACER-L: language; ACER-VE: visuospatial abilities; ACER-OA: orientation/attention; ACER-M: memory; ACER-VF: verbal fluency; Age: patient’s age; Onset: age of onset; TE: time of evolution; NH: number of hospitalizations; SA: number of suicide attempts. Statistically significant correlations are marked with an asterisk.

**Table 5 ijms-26-06372-t005:** Comparison of ACER scores and clinical outcomes according to hs-CRP level.

	Low hs-CRP (n = 73)	High hs-CRP (n = 29)	*p*-Value
ACER-T	85.3 ± 10.04	85.92 ± 7.75	0.76
ACER-L	0.22 ± 0.26	0.25 ± 0.53	0.79
ACER-VE	14.33 ± 1.97	14.65 ± 1.67	0.44
ACER-OA	16.56 ± 1.71	16.62 ± 2.06	0.88
ACER-M	19.87 ± 4.94	19.93 ± 3.57	0.95
ACER-VF	10.52 ± 2.96	10.37 ± 2.51	0.82
NH	3.27 ± 3.89	3.96 ± 4.45	0.44
SA	1.02 ± 1.75	1.82 ± 2.7	0.08

IL-6: interleukin-6; hs-CRP: high-sensitivity C-reactive protein; ACER-T: total; ACER-L: language; ACER-VE: visuospatial abilities; ACER-OA: orientation/attention; ACER-M: memory; ACER-VF: verbal fluency; Age: patient’s age; Onset: age of onset; TE: time of evolution; NH: number of hospitalizations; SA: number of suicide attempts.

**Table 6 ijms-26-06372-t006:** Regression models to predict ACER scores and clinical outcomes from IL-6 and hs-CRP levels.

	ACER-T	ACER-L	ACER-VE	ACER-OA	ACER-M	ACER-VF	NH	SA
Intercept	85.93 (83.8–88)	24.15 (23.7–24.5)	14.54 (14.13–14.95)	16.7 (16.3–17.1)	20 (19–21)	10.53 (9.8–11.1)	3.08 (2.2–3.9)	1.25 (0.76–1.73)
IL-6	−0.107 (−0.16–−0.04) *	−0.021 (−0.03–−0.008) *	−0.02 (−0.035–−0.011) **	−0.019 (−0.031–−0.007) *	−0.03 (−0.06–−0.007) *	−0.005 (−0.024–0.01)	0.05 (0.03–0.08) **	−0.003 (−0.018–0.01)
hs-CRP	1.98 (−2–5.9)	0.69 (−0.12–1.52)	0.34 (−0.44–1.14)	0.19 (−0.58–0.96)	0.74 (−1.25–2.75)	−0.017 (−1.29–1.25)	−0.38 (−2.06–1.29)	0.15 (−0.78–1.09)
Model statistics								
R^2^	0.11	0.11	0.13	0.09	0.06	0.003	0.16	0.003
Adjusted R^2^	0.09	0.1	0.11	0.08	0.04	−0.01	0.14	−0.016
F-statistic	6.4	6.58	7.65	5.45	3.21	0.16	9.38	0.18
*p*-Value	0.002 *	0.002 *	<0.001 *	0.005 *	0.04 *	0.85	<0.001 *	0.83

β-coefficient (95% confidence interval); Statistically significant correlations are marked with an asterisk. IL-6: interleukin-6; hs-CRP: high-sensitivity C-reactive protein; ACER-T: total; ACER-L: language; ACER-VE: visuospatial abilities; ACER-OA: orientation/attention; ACER-M: memory; ACER-VF: verbal fluency; Age: patient’s age; Onset: age of onset; TE: time of evolution; NH: number of hospitalizations; SA: number of suicide attempts. * *p* < 0.05; ** *p* < 0.01.

## Data Availability

Data is contained within the article.

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
