# Peer review of "Inflammation and Cognition in Bipolar Disorder: Diverging Paths of Interleukin-6 and Outcomes"

_ijms, 2025, doi:10.3390/ijms26136372_

Round 1

Reviewer 1 Report

Comments and Suggestions for Authors

The manuscript presents a methodologically sound study evaluating the relationship between inflammatory biomarkers (IL-6 and hs-CRP), neurocognitive functioning, and clinical outcomes in euthymic bipolar disorder patients. There are several areas where the manuscript could be improved to enhance clarity, depth, and generalizability.

1. The sample size (n=109) is modest, and the predominance of women (66%) may limit generalizability, but the lack of control group is a limitation.
2. The mean IL-6 level (8.39 ± 28.4 pg/mL) has a large SD, suggesting a high variability. This should be further analyzed.
3.  The lack of association with hs-CRP contradicts some previous studies.
4. The ACER is a screening tool, supplementing it with domain-specific tests could yield deeper insights.
5. Control for medications, BMI, and physical comorbidities, which may influence inflammation and cognition.
6. The link between IL-6 and hospitalizations is a interesting topic but observational. The author should discuss whether inflammation drives severity or reflects cumulative illness burden.
7. The authors should expand on how IL-6 might directly or indirectly affect cognition.
8. The authors should add a conceptual figure illustrating hypothesized pathways.

Author Response

Response to Reviewers

We sincerely thank the editor and reviewers for their insightful comments, which have significantly helped us improve the quality of our manuscript. We have addressed each comment below, detailing the changes made to the manuscript. Our responses are in red.

The manuscript presents a methodologically sound study evaluating the relationship between inflammatory biomarkers (IL-6 and hs-CRP), neurocognitive functioning, and clinical outcomes in euthymic bipolar disorder patients. There are several areas where the manuscript could be improved to enhance clarity, depth, and generalizability.

We appreciate the comments.

The sample size (n=109) is modest, and the predominance of women (66%) may limit generalizability, but the lack of control group is a limitation.

We appreciate the comments. We have incorporated the low sample size and the disparity between men and women as a limitation. The lack of a control group is already included as a limitation. In this sense, we focused on the concept of “intra-group differences,” emphasizing the clinical heterogeneity that results from the classical clinical categories.

The mean IL-6 level (8.39 ± 28.4 pg/mL) has a large SD, suggesting a high variability. This should be further analyzed.

We appreciate the suggestions. We have incorporated: ““Another relevant aspect to consider is that the average IL-6 had a large dispersion in our sample. This may be due to different phenomena beyond psychiatric diagnosis, such as age, sex, genetic variants involved in IL-6 signaling, concomitant diseases, among others [45]. These factors were not considered in the adjustment of the proposed models, a fact that constitutes a limitation of this study.” We have included a new reference: Coe et al.

The lack of association with hs-CRP contradicts some previous studies.

We appreciate the comment. Indeed, the lack of association between hs-CRP and neurocognitive and clinical outcomes is in the opposite direction to previous studies. It was addressed in the Introduction. In the Discussion section, we have incorporated: “This lack of association may be partially explained due to the normal hs-CRP levels in our sample, but it is important to consider that IL-6 directly reflects the activity of cytokines, which are the main mediators and regulators of the inflammatory response, while hs-CRP is a more global marker of the acute phase response, but does not indicate the specific cytokine or molecular pathways involved.”

The ACER is a screening tool, supplementing it with domain-specific tests could yield deeper insights.

We appreciate the comments and agree with them. It is not an easy task to find a more specific instrument for an adequate measurement of neurocognition in BD, therefore, we have leaned towards the ACER as it offers a multidimensional overview of the many planes that compose neurocognition. In this regard, we have included: “Regarding the measurement of neurocognition, ACER offers a first global assessment, so other instruments can complement a more in-depth evaluation of this construct.”

Control for medications, BMI, and physical comorbidities, which may influence inflammation and cognition.

We appreciate the comments and agree with them. These aspects were included as limitations. Unfortunately, we do not have those data.

The link between IL-6 and hospitalizations is a interesting topic but observational. The author should discuss whether inflammation drives severity or reflects cumulative illness burden.

We appreciate the suggestion. We have included: “Inflammatory burden has been clearly recognized as a preponderant factor in immune-mediated diseases [62], but in psychiatric conditions this relationship has been scarcely studied. In unipolar depression, a correlation between inflammatory burden and depressive outcomes has been corroborated [63], including hospital variables such as that observed in our results. In BD, Queissner et al. [64] concluded that individuals who were exposed to a higher level of inflammation over time suffered from more manic symptoms in this period. This finding allows to indirectly hypothesize that people with higher levels of inflammation would require a greater NH. However, the relationship between inflammation and NH is still statistical and requires further theoretical understanding.” We have incorporated the references: Wu et al., Shao et al., and Queissner et al.

The authors should expand on how IL-6 might directly or indirectly affect cognition.

We appreciate the suggestion. We have deepened the explanation about the effects of IL-6 on cognition: As mentioned above, the proposed mechanisms by which IL-6 levels alter neurocognitive functioning are multiple. Recently, studies of resting-state functional connectivity magnetic resonance imaging have suggested that immune dysregulation is involved in connectivity abnormalities in limbic and somatomotor networks that account for the neurocognitive alterations in BD. These alterations include structural modifications, such as reduced cortical thickness, subcortical volumes [59]. At the cellular level, IL-6 is known to modulate the process of neurogenesis throughout neurodevelopment. However, prolonged exposure at high levels may lead to reduced proliferation and increased neuronal apoptosis. This would be one of the main mechanisms underlying the alterations that occur in the aforementioned structures, all of which are involved in cognitive processes. However, the conclusions on the effects of IL-6 at the cellular level have been obtained from in vitro studies and in animal models; therefore, it is necessary to deepen research in this area [60].” We have incorporated two new references: Altamura et al. and Bradburn et al.

The authors should add a conceptual figure illustrating hypothesized pathways.

We appreciate the suggestion. We have included a figure which synthesizes our main results and some proposed mechanisms (Figure 1).

Reviewer 2 Report

Comments and Suggestions for Authors

The manuscript presents valuable findings on the role of IL-6 in BD-related cognitive dysfunction and hospitalizations. Addressing the above points—particularly on limitations, clinical implications, and methodological clarity—would enhance its impact. 

1. The study provides valuable insights into the association between IL-6 and neurocognitive functioning in BD. However, the lack of a control group limits the ability to generalize findings. Consider acknowledging this limitation more explicitly and discussing its implications.

2. The manuscript lacks details on the inclusion/exclusion criteria for participants (e.g., medication use, comorbidities). Clarifying these would improve reproducibility.

3. Provide more information on the ELISA and immunoturbidimetry methods (e.g., intra- and inter-assay coefficients of variation) to ensure transparency.

4. Discuss the clinical implications of your findings more explicitly. For example, could IL-6 be a target for interventions to improve cognitive outcomes in BD?

5. It will be a great idea to clarify how euthymia was defined (e.g., specific cutoff scores on the Young Mania Rating Scale and Hamilton Depression Rating Scale)

6. Consider adding sensitivity analyses if possible to account for potential confounders (e.g., age, medication use).

7. The mean IL-6 level (8.39 ± 28.4 pg/mL) has a large standard deviation. Discuss potential reasons (e.g., outliers, skewed distribution) and how this was addressed.In addition, Ensure consistency in formatting (e.g., decimal places, significance indicators).

8. The models explain ~10% of the variance. Discuss whether this effect size is clinically meaningful.

Author Response

Response to Reviewers

We sincerely thank the editor and reviewers for their insightful comments, which have significantly helped us improve the quality of our manuscript. We have addressed each comment below, detailing the changes made to the manuscript. Our responses are in red.

The manuscript presents valuable findings on the role of IL-6 in BD-related cognitive dysfunction and hospitalizations. Addressing the above points—particularly on limitations, clinical implications, and methodological clarity—would enhance its impact. 

We appreciate the comments and suggestions.

The study provides valuable insights into the association between IL-6 and neurocognitive functioning in BD. However, the lack of a control group limits the ability to generalize findings. Consider acknowledging this limitation more explicitly and discussing its implications.

We appreciate the comments. The lack of a control group is already included as a limitation. In this sense, we focused on the concept of “intra-group differences,” emphasizing the clinical heterogeneity that results from the classical clinical categories. We have incorporated: “We did not incorporate a control group constituted by participants without BD, an aspect to be considered in future designs; however, we focused on the concept of “intra-group differences,” emphasizing the clinical heterogeneity that results from the classical clinical categories.”

The manuscript lacks details on the inclusion/exclusion criteria for participants (e.g., medication use, comorbidities). Clarifying these would improve reproducibility.

We appreciate the comment. We have incorporated: “We did not exclude patients with medical comorbidities or use of specific drugs.” It was assumed as a limitation: “We also did not have measurements of inflammation and cognition in active mood phases of the disease, nor did we control for factors that may have modified the results, such as the use of antipsychotics, the presence of chronic diseases related to inflammation, and body weight.”

Provide more information on the ELISA and immunoturbidimetry methods (e.g., intra- and inter-assay coefficients of variation) to ensure transparency.

We appreciate the suggestion. We have modified the inflammatory analysis section:

“4.5. Inflammatory variables

Inflammatory analyses were conducted at the Toxicology Laboratory Universidad de Valparaíso. Quantitative determination of IL-6 and hs-CRP in serum was determined from peripheral venous blood samples (forearm vein) by standard venipuncture. All blood samples were collected between 07:00 and 09.00 AM into vacuum blood-collecting tubes containing anticoagulant ethylenediaminetetraacetic acid (EDTA) 1 mg/mL, and the samples were obtained by centrifugation immediately for 10 minutes at 3500 rpm.

hs-CRP

The determination and quantification of hs-CRP was performed in a BN ProSpec® automated equipment and the reagent used for this measurement was CardioPhase® hs-CRP. The technique used was immunonephelometry with intensifying particles, which consists of the use of polystyrene particles coated with a specific monoclonal antibody against human CRP, which when mixed with samples containing CRP form aggregates that scatter the incident light beam. The intensity of the scattered light depends on the concentration of the corresponding protein in the sample. The final titration is made by comparison with a standard curve of known concentration, the lower limit of this curve being the sensitivity (LOD) of the assay, which typically corresponds to 0.0175 mg/dL. The following reference values were used according to the manufacturer CardioPhase hs-CRP in BN ProSpec equipment: < 0.10 mg/L (low risk), 0.10-0.30 mg/L (moderate risk), > 0.30 mg/L (high risk), and > 1.00 mg/L (very high risk).

IL-6

Quantitative determination of IL-6 was performed through quantitative enzyme-linked immunosorbent assay (ELISA) using a commercially available kit (Quantikine, R&D Systems, Minneapolis, USA) taking as reference value IL-6 < 3.5 pg/mL according to the manufacturer instructions. Briefly, this assay employs the quantitative sandwich enzyme immunoassay technique, whose microplate has been pre-coated with a monoclonal antibody specific for human IL-6. 100 μL of standards or plasma simple was added to each well and incubated for 2 hours at RT and any IL-6 present is bound by the immobilized antibody. To construct a calibration curve, we diluted Human IL-6 Standard with 1X Assay diluent to 300, 100, 50, 25, 12.5, 6.25, 3.13 and 0 pg/mL and used them as standard samples. After washing away any unbound substances, 200 μL of human IL-6 conjugate (enzyme-linked polyclonal antibody specific for human IL-6) was added to the wells followed by a 2 hours incubation at room temperature. Then, 200 μL of substrate solution was added to each well and incubated for 20 minutes at RT and color develops in proportion to the amount of IL-6 bound. The color development is stopped with 50 μL of stop solution and the intensity of the color is measured. Absorbance was interpolated from a calibration curve. The sensitivity was 0.7 pg/mL in a range of 0-300 pg/mL with intra-assay and inter-assay coefficients from 4.2% to 7%, respectively. There were no undetectable values. All samples were over the detection limit.

Discuss the clinical implications of your findings more explicitly. For example, could IL-6 be a target for interventions to improve cognitive outcomes in BD?

We appreciate the comment. We have incorporated: “Despite its biological plausibility, the clinical utility of pharmacological targets that seek to modulate the IL-6-mediated inflammatory pathway is a matter of debate. To date there are few clinical trials with highly variable results regarding the pro-cognitive effect of IL-6 modulators; most of these studies have been conducted in samples of patients with Alzheimer's disease [67], making direct extrapolation to our population of interest difficult. In parallel, other interventions of a non-pharmacological nature such as dietary interventions have received some degree of interest as modulators of the IL-6-related inflammatory pathway, but evidence is still scarce [68].” We have incorporated references [67] and [68].

It will be a great idea to clarify how euthymia was defined (e.g., specific cutoff scores on the Young Mania Rating Scale and Hamilton Depression Rating Scale)

We appreciate the suggestion. We have incorporated the cutoff scores for both scales (≤ 6 points).

Consider adding sensitivity analyses if possible to account for potential confounders (e.g., age, medication use).

We appreciate the comments and agree with them. These aspects were included as limitations. Unfortunately, we do not have complete data on medication use. Due to the anti-inflammatory effect of some drugs commonly used in BD, this analysis would have been of great relevance.

The mean IL-6 level (8.39 ± 28.4 pg/mL) has a large standard deviation. Discuss potential reasons (e.g., outliers, skewed distribution) and how this was addressed. In addition, Ensure consistency in formatting (e.g., decimal places, significance indicators).

We appreciate the suggestions. We have incorporated: ““Another relevant aspect to consider is that the average IL-6 had a large dispersion in our sample. This may be due to different phenomena beyond psychiatric diagnosis, such as age, sex, genetic variants involved in IL-6 signaling, concomitant diseases, among others [45]. These factors were not considered in the adjustment of the proposed models, a fact that constitutes a limitation of this study.” We have included a new reference: Coe et al.

The models explain ~10% of the variance. Discuss whether this effect size is clinically meaningful.

We appreciate the comments and agree with them. We have incorporated it as a limitation: On the other hand, the regression models explained only a small proportion of the variance of the cognitive and clinical outcomes, which could be since other relevant variables were not considered, especially clinical variables. Nevertheless, there is sufficient evidence from in vitro analyses, in vivo models, clinical studies, and meta-analyses to support that inflammation has a significant impact on clinically meaningful outcomes”.

Round 2

Reviewer 1 Report

Comments and Suggestions for Authors

The authors revised the manuscript well.